# Joint effect of alcohol drinking and tobacco smoking on all-cause mortality and premature death in China: A cohort study

Zhang Hongli[1,2]☯, Xueyuan Bi[3]☯, Nanbo Zheng[2], Chao Li[1], Kangkang Yan●[4]*

**1** Xi'an Jiaotong University Health Science Center, Xi'an, China, **2** Department of Pharmacy, Xi'an Central Hospital, Xi'an, China, **3** Department of Pharmacy, Xi'an Honghui Hospital, Xi'an, China, **4** Department of Pharmacy, Xi'an No.3 Hospital, the Affiliated Hospital of Northwest University, Xi'an, China

☯ These authors contributed equally to this work.
* kky_hospital@163.com

**Data Availability Statement:** All datasets are available from the CHARLS database (charls.pku.edu.cn/pages/data/111/en.html; dataset titles used in this study were 2013 CHARLS Wave2 and 2011

## Abstract

### Background

Tobacco smoking and alcohol drinking are associated with several diseases, and studies on the joint effects of smoking and drinking are rare.

### Objective

This study investigates the joint effects of tobacco smoking and alcohol drinking on all-cause and premature mortality in a contemporary cohort.

### Methods

The China Health and Retirement Longitudinal Study (CHARLS) is an ongoing nationally representative survey of subjects aged over 45 years in China that was performed every two years for a total of three waves from 2011 to 2015 in China. We used weighted logistic regression models to estimate the joint effects of tobacco smoking and alcohol drinking on all-cause and premature mortality.

### Results

After adjusting for prespecified confounders, the odds ratios (ORs) of all-cause mortality were 1.51 (95% CI: 1.09–2.10) and 1.47 (95% CI: 1.03–2.08) in smokers and smokers/drinkers, respectively. Compared with nonsmokers/nondrinkers, the OR of smokers/drinkers for premature death was 3.14 (95% CI: 1.56–6.34). In the female subgroup, there was an approximately 5-fold (OR = 4.95; 95% CI: 2.00–12.27) odds of premature mortality for smokers/drinkers compared to nonsmokers/nondrinkers.

### Conclusion

This study found a joint effect of tobacco smoking and alcohol drinking on all-cause and premature mortality among a contemporary and nationally representative cohort in China. Our

CHARLS Wave1). The authors of the present study had no special privileges in accessing these datasets which other interested researchers would not have.

**Funding:** The authors received no specific funding for this work.

**Competing interests:** No authors have competing interests.

results suggested that the joint effects were more pronounced in women, but further research is needed.

## Introduction

Tobacco smoking is a well-established risk factor for many diseases, including stroke and coronary heart disease [1]. Additionally, several studies, including the British Doctors Study [2], the American Cancer Society Cancer Prevention Study [3] and a pooled analysis of Asian cohorts [4], have shown that tobacco smoking is associated with an increased risk of mortality. It has been reported that tobacco smoking is the second leading risk of mortality worldwide [5,6] and the first leading risk of mortality among Chinese men [7].

Alcohol drinking is another leading risk factor for mortality and disability [8]. Although some studies reported a J-shaped association between alcohol drinking and all-cause mortality, which suggested that low alcohol use may confer some degree of protection [9,10], this view was challenged by recent Mendelian randomization studies and meta-analyses [8,11,12]. These protective associations between moderate alcohol drinking and health may, however, be attributed to the effect of biases (e.g., from study design, confounders, selection of participants) [12,13].

Although the effect of tobacco smoking and alcohol drinking on death has been well documented, studies on the joint effects of these two leading risk factors on all-cause or premature mortality are rare. Therefore, this study aimed to estimate the joint effects of tobacco smoking and alcohol drinking on all-cause and premature mortality by using a nationally representative cohort.

## Materials and methods

### Study population

Data for this study were drawn from the China Health and Retirement Longitudinal Study (CHARLS). The baseline survey of this study was conducted in 2011–2012 and involved 17 708 subjects. The details of the study design have been reported previously [14,15]. In summary, it is an ongoing nationally representative survey that was performed every two years for a total of 3 waves from 2011 to 2015 in China. Adults aged 45 years or older living in China were enrolled in the baseline survey and followed up in waves 2, 3 and 4. Subjects were chosen with a multistage stratified sampling design in 450 villages, 150 counties and 28 provinces to ensure the representativeness of the sample and can be weighted to obtain national estimates.

### Data collection and variable definition

Each subject was administered a survey interview that included information on date of birth, sex, educational level (illiterate, primary, secondary/high school, college/university or above), marital status (married/divorced/widowed, unmarried), residence status (rural, urban), tobacco smoking habits, alcohol drinking, height and weight, medicine usage and medical history. Blood was also taken for laboratory testing. We defined the history of cardiovascular disease (CVD) if subjects reported having a history of stroke, heart failure or CHD. Overweight was defined as $24 \, \text{kg/m}^2 \leq \text{BMI} < 28 \, \text{kg/m}^2$, and obesity was defined as a $\text{BMI} \geq 28 \, \text{kg/m}^2$. Blood pressure was measured by an Omron model HEM-7200. The SBP and DBP were calculated by averaging the available 3 blood pressure measurements (approximately 45 s apart) for

each subject. We defined hypertension as an SBP of 140 mm Hg or more, a DBP of 90 mm Hg or more, or self-reported use of antihypertensive treatment. Diabetes was defined as hemoglobin $A_{1C} \geq 6.5\%$, use of anti-diabetic medication, or self-reported history of diabetes. Dyslipidemia was defined as the use of dyslipidemia medication or having one of the following conditions: total cholesterol $\geq 6.19$ mmol/L, low density lipoprotein cholesterol $\geq 4.14$ mmol/L, or triglycerides $\geq 2.27$ mmol/L. Subjects were classified as tobacco smokers if they had ever smoked more than 100 cigarettes in their lives. An alcohol drinker was defined as a participant drinking alcoholic beverages at least once a month. Participants were further divided into four groups: nonsmoker/nondrinker, drinker, smoker, and smoker/drinker.

Participant status (dead or alive) was recorded in each wave; however, the death date was only recorded in wave 2 (2013 follow-up). The follow-up rate for the baseline survey (2011) was 80.5%, 69% in urban areas and 94% in rural areas, lower follow-up rate in the year of 2013 in urban areas as is common in most surveys conducted in developing countries. We calculated the age of death from the interval between the date of birth and the death date. In our current analysis, we defined premature death according to the average life expectancy in China in 2011, which died before 72.7 years in men and 76.9 years in women [16]. Detailed information about the data quality management has been estimated previously [17]. Since we were performing secondary analyses of deidentified data, we sought no approval from institutional review boards.

## Statistical analysis

Summary statistics for the baseline characteristics and risk factors by sex were computed as the proportions or means and compared across different smoking and drinking status groups by using ANOVA or a chi-square test. Baseline characteristics were also compared between participants with and without missing data. The joint effects of tobacco smoking and alcohol drinking on all-cause and premature mortality were estimated using weighted logistic regression models. Unadjusted and adjusted odds ratios (ORs) and their 95% confidence intervals (CIs) were obtained. The multivariable models were adjusted for sex, age, educational level, current residence status, marital status, hypertension, dyslipidemia, diabetes, history of CVD, and overweight or obesity. All reported $P$ values were 2-tailed, and $< 0.05$ was considered statistically significant. Analyses were performed with SAS software version 9.4.

## Results

A total of 17708 individual participants were interviewed in the CHARLS, and 17251 participants who were aged 45 years or more were eligible for the current analysis. Of those, 171 participants were excluded due to missing age/sex (n = 38), tobacco smoking (n = 126) or alcohol drinking status (n = 7). The remaining 17080 participants are included in the current analysis. Baseline characteristics were comparable between participants with and without missing data, and most of the variables were non-significant (S3 Table). A total of 26.8% (4583/17080) of participants were both cigarette smokers and alcohol drinkers, with a substantially higher prevalence in male subjects (52.7% [4391/8332] in men and 2.19% [192/8748] in women). The mean age of participants among different smoking and drinking groups ranged from 50.3 to 60.9 years, and most participants were from rural areas and married. The prevalence of overweight or obesity was higher in the nonsmoker/nondrinker group. The prevalence of other cardiovascular-related risk factors (hypertension, dyslipidemia, diabetes, history of CVD) was similar among the four different smoking and drinking groups (Table 1).

Fig 1A and 1B and S1 Table shows the joint effect of smoking and drinking on all-cause mortality. Generally, the odds of all-cause mortality were higher in the smoker group

**Table 1. Baseline characteristics among different smoking and drinking groups by sex.**

| | Nonsmoker/Nondrinker | Drinker | Smoker | Smoker/Drinker | P |
|---|---|---|---|---|---|
| **All participants** | | | | | |
| Number of subjects | 8029 | 2159 | 2309 | 4583 | |
| Age (years) | 50.30 (10.06) | 59.26 (9.83) | 60.89 (10.03) | 59.57 (9.37) | <0.001 |
| Age group | | | | | <0.001 |
| <50 | 1798 (22.39) | 483 (22.37) | 395 (17.11) | 868 (18.94) | |
| 50–59 | 2885 (35.93) | 754 (34.92) | 795 (34.43) | 1714 (37.40) | |
| 60–69 | 2077 (25.87) | 599 (27.74) | 636 (27.54) | 1321 (28.82) | |
| 70–79 | 954 (11.88) | 247 (11.44) | 392 (16.98) | 555 (12.11) | |
| ≥80 | 315 (3.92) | 76 (3.52) | 91 (3.94) | 125 (2.73) | |
| Middle school education (%) | 2109 (26.32) | 669 (31.00) | 738 (31.99) | 1752 (38.28) | |
| Rural (%) | 6236 (77.75) | 1656 (76.74) | 1839 (79.71) | 3603 (78.65) | 0.064 |
| Married (%) | 6838 (85.17) | 1896 (87.86) | 1966 (85.15) | 4152 (90.60) | <0.001 |
| Hypertension (%) | 3159 (39.34) | 834 (38.63) | 810 (35.08) | 1764 (38.49) | 0.003 |
| Dyslipidemia (%) | 2630 (32.76) | 659 (30.52) | 770 (33.35) | 1359 (29.65) | 0.001 |
| Diabetes (%) | 653 (8.13) | 178 (8.24) | 167 (7.23) | 288 (6.28) | 0.001 |
| Overweight or obesity (%)* | 2886 (46.29) | 712 (42.61) | 588 (32.15) | 1118 (31.98) | <0.001 |
| History of CVD (%) | 1211 (15.08) | 271 (12.55) | 333 (14.42) | 556 (12.13) | <0.001 |
| **Male** | | | | | |
| Number of subjects | 1091 | 1083 | 1767 | 4391 | |
| Age (years) | 61.42 (10.47) | 59.20 (9.96) | 60.17 (9.75) | 59.37 (9.26) | <0.001 |
| Age group | | | | | <0.001 |
| <50 | 198 (18.15) | 253 (23.36) | 319 (18.05) | 855 (19.47) | |
| 50–59 | 332 (30.43) | 367 (33.89) | 638 (36.11) | 1655 (37.69) | |
| 60–69 | 308 (28.23) | 299 (27.61) | 490 (27.73) | 1254 (28.56) | |
| 70–79 | 202 (18.52) | 124 (11.45) | 264 (14.94) | 517 (11.77) | |
| ≥80 | 51 (4.67) | 40 (3.69) | 56 (3.17) | 110 (2.51) | |
| Middle school education (%) | 428 (39.34) | 460 (42.51) | 668 (37.83) | 1724 (39.32) | <0.001 |
| Rural (%) | 777 (71.22) | 772 (71.35) | 1390 (78.75) | 3441 (78.38) | <0.001 |
| Married (%) | 978 (89.64) | 1003(92.70) | 1563(88.46) | 4008(91.28) | <0.001 |
| Hypertension (%) | 402 (36.85) | 425 (39.24) | 570 (32.26) | 1671 (38.06) | <0.001 |
| Dyslipidemia (%) | 356 (32.63) | 333 (30.75) | 573 (32.43) | 1300 (29.61) | 0.077 |
| Diabetes (%) | 76 (6.97) | 95 (8.77) | 116 (6.56) | 275 (6.26) | 0.031 |
| Overweight or obesity (%)* | 300 (38.56) | 342 (42.59) | 408 (29.76) | 1058 (31.63) | <0.001 |
| History of CVD (%) | 139 (12.74) | 143 (13.20) | 197 (11.15) | 511 (11.64) | 0.292 |
| **Female** | | | | | |
| Number of subjects | 6938 | 1076 | 542 | 192 | |
| Age (years) | 58.97 (9.96) | 59.32 (9.70) | 63.22 (10.56) | 64.30 (10.49) | <0.001 |
| Age group | | | | | <0.001 |
| <50 | 1600 (23.06) | 230 (21.38) | 76 (14.02) | 13 (6.77) | |
| 50–59 | 2553 (36.80) | 387 (35.97) | 157 (28.97) | 59 (30.73) | |
| 60–69 | 1769 (25.50) | 300 (27.88) | 146 (26.94) | 67 (34.90) | |
| 70–79 | 752 (10.84) | 123 (11.43) | 128 (23.62) | 38 (19.79) | |
| ≥80 | 264 (3.81) | 36 (3.35) | 35 (6.46) | 15 (7.81) | |
| Middle school education (%) | 1681 (24.27) | 209 (19.42) | 70 (12.94) | 28 (14.58) | <0.001 |
| Rural (%) | 5459 (78.77) | 884 (82.16) | 449 (82.84) | 162 (84.82) | 0.003 |
| Married (%) | 5860 (84.46) | 893 (82.99) | 403 (74.35) | 144 (75.00) | <0.001 |
| Hypertension (%) | 2757 (39.74) | 409 (38.01) | 240 (44.28) | 93 (48.44) | 0.008 |

*(Continued)*

**Table 1.** (Continued)

| | Nonsmoker/Nondrinker | Drinker | Smoker | Smoker/Drinker | P |
|---|---|---|---|---|---|
| Dyslipidemia (%) | 2274 (32.78) | 326 (30.30) | 197 (36.35) | 59 (30.73) | 0.092 |
| Diabetes (%) | 577 (8.32) | 83 (7.71) | 51 (9.41) | 13 (6.77) | 0.582 |
| Overweight or obesity (%)* | 2586 (47.39) | 370 (42.63) | 180 (39.30) | 60 (39.74) | <0.001 |
| History of CVD (%) | 1072 (15.45) | 128 (11.90) | 136 (25.09) | 45 (23.44) | <0.001 |

*missing N = 2035 for male and N = 1814 for female.

(unadjusted OR = 1.75 95% CI: 1.39, 2.19). Similar results were found in the subgroup analysis; smoking in females was associated with a 2.3-fold (unadjusted analysis: OR = 2.34; 95% CI: 1.60–3.43) odds of all-cause mortality. Compared with nonsmokers, the odds of all-cause mortality were higher in smokers/drinkers, and smokers/drinkers increased the odds of all-cause mortality with an OR value equal to 1.75 (95% CI: 1.39–2.19, unadjusted analysis). In the subgroup analysis of females, the OR value was 3.14 (95% CI: 1.82–5.14, unadjusted analysis). (Fig 1A, S1 Table) In multivariable regression, after controlling for confounders, the associations remained significant. The OR values of all-cause mortality were 1.47 (95% CI: 1.03–2.08) and 1.51 (95% CI: 1.09–2.10) in smokers and smokers/drinkers, respectively. In the subgroup analysis, smokers (OR = 1.56; 95% CI: 0.99–2.44) and smokers/drinkers (OR = 2.02; 95% CI: 0.95–4.30) in females were associated with an increase in the odds of all-cause mortality compared with nonsmokers/nondrinkers in females. Although the associations were not significant after controlling for confounders, the lower limits of the 95% CI were close to 1. (Fig 1B, S1 Table).

Compared with nonsmokers and nondrinkers, smokers/drinkers were associated with an increase in the odds of premature death in the unadjusted analysis (OR = 2.12; 95% CI: 1.46–3.08). After controlling for confounders, the OR value was 3.14 (95% CI: 1.56–6.34). The odds of premature death were 2.2-fold (adjusted analysis: OR = 2.17; 95% CI: 1.11–4.22) higher in smokers. In the unadjusted analysis, the OR of premature death in smokers was 1.49 (95% CI: 0.90–2.45). In the subgroup analysis, female smoking/drinking was associated with a 5-fold (unadjusted analysis: OR = 5.36; 95% CI: 2.32–12.36; adjusted analysis: OR = 4.95; 95% CI: 2.00–12.27) increase in the odds of premature death. The odds of all-cause and premature

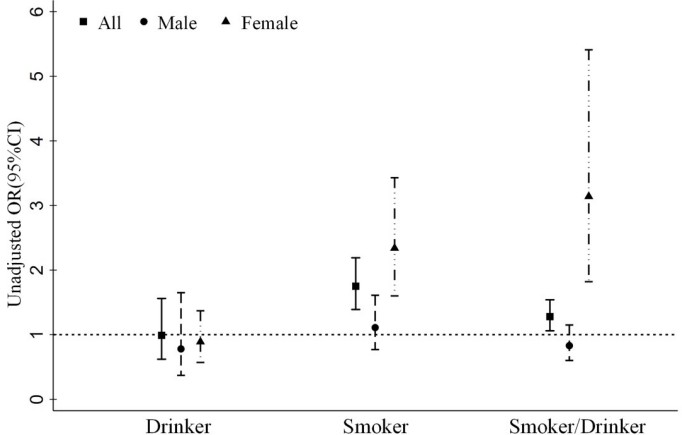 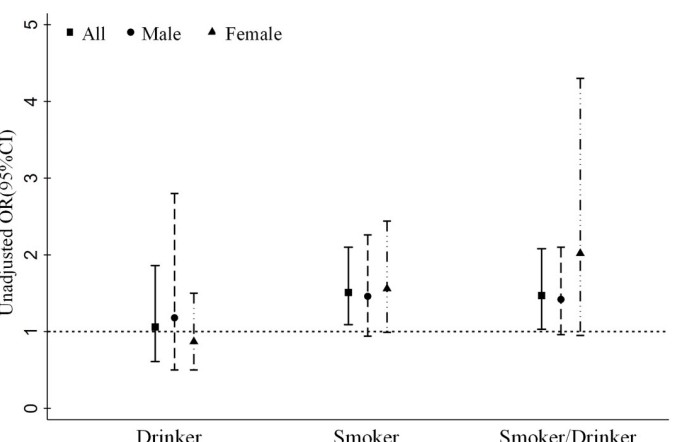

**Fig 1.** A. Unadjusted odds ratio of all-cause mortality among different smoking and drinking groups by sex. B. Adjusted odds ratio of all-cause mortality among different smoking and drinking groups by sex (Adjusted variables include sex, age, middle school education, residence status, marital status, hypertension, dyslipidemia, diabetes, history of CVD, overweight or obesity).

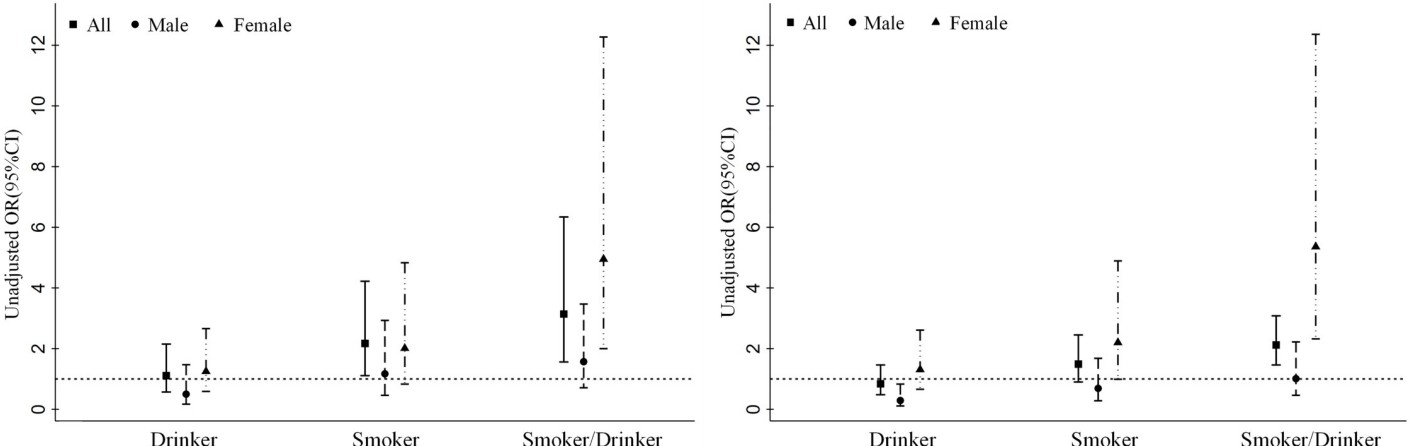

**Fig 2.** A. Unadjusted odds ratio of premature death among different smoking and drinking groups by sex. B. Adjusted odds ratio of premature death among different smoking and drinking groups by sex (Adjusted variables include sex, age, middle school education, residence status, marital status, hypertension, dyslipidemia, diabetes, history of CVD, overweight or obesity).

mortality were similar and not significant between the nonsmoker/nondrinker group and the drinker group. (Fig 2A and 2B and S2 Table).

## Discussion

In our present analysis, we found that alcohol drinking could strengthen the harmful effects of tobacco smoking. Our study indicates that compared with nonsmokers/nondrinkers, smokers and drinkers have 75% higher risks of all-cause mortality or 112% higher risk of premature mortality. The study also indicated that women are likely to have a higher risk of all-cause or premature mortality from the joint effects of tobacco smoking and alcohol drinking than men.

Alcohol drinking has been reported to increase the risk of all-cause mortality with a U- or J-shaped relationship. This indicated a lower risk of mortality among subjects with light to moderate drinking and a higher risk of death among those who are heavy drinkers [9,10]. However, a study from a large cohort showed that the beneficial effects of light and moderate alcohol use were offset by tobacco smoking, and the risk of all-cause mortality increased with smoking intensity from 0.8 (95% CI: 0.6, 1.0) for nonsmokers to 1.0 (0.9, 1.2) for moderate smokers and 1.4 (95% CI: 1.2, 1.7) for heavy smokers [18]. The negative combined effects of smoking and drinking were also reported [19,20]. Our study was consistent with these findings and found that people who are both smokers and drinkers have a higher risk of all-cause mortality.

The premature death of a middle-aged person is often devastating to a household and is often associated with decreased income and mortality in the affected families. Identification of the associated risk is crucial to health planning and policy development [21]. In our study, we also tried to explore the adverse association of this synergistic effect of tobacco smoking and alcohol drinking with premature death. Consistent with the results for all-cause mortality, we found that compared with nonsmokers and nondrinkers, smokers and drinkers were significantly associated with early death, despite the nonsignificance among subgroups by sex due to the small sample size.

It is noteworthy that the sex difference was found in both all-cause mortality and premature death. It has been noted that female smokers and/or drinkers are more likely to have greater mortality risk. This phenomenon has been reported in previous publications [1,4,9,19,22,23]

assessing smoking or drinking risk, even among those with existing diseases such as diabetes [24]. The mechanisms underlying the sex difference in mortality may be attributed to their biological or related to differences in smoking behavior between men and women. It seems that men are likely to be exposed to more risks than women, which may confer the risk from smoking or drinking use. This can be derived from our data. In our study, we found that 52.7% of men were both cigarette smokers and alcohol drinkers, whereas only 2.19% were found in women.

Several limitations of the study should be mentioned. First, due to the limited information from this public database, further analysis by drinking amount (the definition of at least once a month of drinking was the only and the largest of frequencies of drinking option) or the cause of death, such as CVD mortality or cancer mortality, is not available. Several studies have shown that the effect of alcohol drinking on mortality is generally found to depend on the amount consumed and presented differently by cause-specific mortality. However, this was argued by other studies suggesting that the nonlinear relationship is explained by the study design or selected bias across age-sex strata [11,12]. Second, this study is an ongoing study, and the follow-up period is relatively short. Combined with the relatively small sample of death events that have been followed, the statistical power is limited for subgroup analysis and sensitivity analysis after excluding individuals who had events in the first five or ten years of follow-up to avoid possible inverse causal relationships. Third, as the date of death was only released for data from the 2013 follow-up survey, information on age of death and premature death can only be obtained for this survey.

In summary, our present analysis indicated that alcohol drinking could further increase the risk of total death or premature death from smoking. Further studies are needed to confirm the higher risk of death from the joint effect of smoking and drinking among the female population.

## Supporting information

**S1 Table. Odds ratio of all-cause mortality among different smoking and drinking groups by sex.**
(DOCX)

**S2 Table. Odds ratio of premature death among different smoking and drinking groups by sex.**
(DOCX)

**S3 Table. Comparison of baseline characteristics between participants with and without missing data.**
(DOCX)

**S4 Table. Name of variables and datasets of CHARLS data used.**
(DOCX)

**S1 File.**
(SAS)

**S2 File.**
(PDF)

## Acknowledgments

We would like to thank the CHARLS (China Health and Retirement Longitudinal Study) participants and trial investigators. This article was prepared using research materials obtained from the Peking University Open Research Data Platform.

## Author Contributions

**Conceptualization:** Kangkang Yan.

**Data curation:** Xueyuan Bi.

**Formal analysis:** Zhang Hongli, Kangkang Yan.

**Writing – original draft:** Zhang Hongli, Xueyuan Bi.

**Writing – review & editing:** Nanbo Zheng, Chao Li, Kangkang Yan.

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
