## [Decision Letter · Decision Letter 0]

20 Oct 2020

PONE-D-20-20926

Joint effect of alcohol use and tobacco smoke on all-cause mortality and premature death in China: a cohort study

PLOS ONE

Dear Dr. Yan,

Thank you for submitting your manuscript to PLOS ONE. After careful consideration, we feel that it has merit but does not fully meet PLOS ONE’s publication criteria as it currently stands. Therefore, we invite you to submit a revised version of the manuscript that addresses the points raised during the review process.

Please, address the comments from both reviewers for further consideration.

We look forward to receiving your revised manuscript.

Kind regards,

Associate Professor Dr Muhammad Aziz Rahman,

MBBS, MPH, CertGTC, GCHECTL, PhD

Academic Editor

PLOS ONE

Journal Requirements:

2. Please refer to any post-hoc corrections to correct for multiple comparisons during your statistical analyses. If these were not performed please justify the reasons. Please refer to our statistical reporting guidelines for assistance (https://journals.plos.org/plosone/s/submission-guidelines.#loc-statistical-reporting).

3.Thank you for stating the following financial disclosure:

 [No].

Reviewers' comments:

Reviewer's Responses to Questions

**Comments to the Author**

1. Is the manuscript technically sound, and do the data support the conclusions?

Reviewer #1: Partly

Reviewer #2: Partly

2. Has the statistical analysis been performed appropriately and rigorously? 

Reviewer #1: Yes

Reviewer #2: I Don't Know

3. Have the authors made all data underlying the findings in their manuscript fully available?

Reviewer #1: Yes

Reviewer #2: Yes

4. Is the manuscript presented in an intelligible fashion and written in standard English?

Reviewer #1: No

Reviewer #2: No

5. Review Comments to the Author

Reviewer #1: This is an interesting paper on the combined effect of alcohol consumption and tobacco use on mortality and premature death in China.

However, I see some major limitations to the paper. I see in your discussion section you have mentioned that the paper is limited by not having information on the amount that participants drunk. This is a major limitation. And the definition of at least once a month seems small. Having moderate, heavy drinking would really add to the paper. Is there some other way of modelling this from other studies/datasets? Also not having information on cause of mortality limits the results of this paper significantly.

Are you able to look at different levels of tobacco use and drinking? Currently the paper only looks at "ever" smoking, 100 cigarettes in their lifetime. It would useful also to look smoking frequency. Is this possible?

The manuscript requires a major edit for grammar. I would note some down but without line numbers it is very difficult to do this.

The term tobacco smoke, isn't quite right. Tobacco smoking is more suitable in most places. The background sentence in the abstract doesn't make sense. Something like "Tobacco smoking and alcohol use have been shown to be associated with several diseases, however few studies have looked at the combined effect of smoking and drinking" may be more suitable.

Abstract in the results you don't need a "the" before confounders.

I would use "sex" rather than "gender" in the manuscript and keep this consistent throughout.

I would find it easier to call the groups: non-smoker/non-drinker, smoker, drinker, smoker/drinker. Define it earlier on, but it is confusing to read as it is.

Discussion, third line down you say non-smoker twice.

Reviewer #2: Be more careful with phrasings that insinuate causality.

Present study is representative cohort design. Due to nature of observational study, the results did not showed causality Causal language (including use of terms such as "effect," "efficacy," "cause," and "x increased y") should be used only for randomized trials. For all other study designs (including meta-analyses of randomized trials), methods and results should be described in terms of association or correlation (eg, "x associated with an increase in y") and should avoid cause-and-effect wording.

Recent, the 4th wave of CHARLS (2018) has been released. The data can be updated using 4 waves from 2011 to 2018.

This study have many language/grammar errors. For example, in the section of results of abstract, ’smoker and drinker increased the odds of premature death’.

Please add the definition of outcome in the section of methods.( Premature death defined as mortality before age 72.7 years in men and 76.9 years in women, which were the average life expectancies in China in 2011).

The authors stated that they used weighted logistic regression. I am looking forward to the code for validating their correctness of analysis.

More sensitivity analysis need to be conducted to show the robustness of results. For example, excluding individuals who had events in the first five years of follow-up to avoid possible inverse causal relationships and excluding individuals who had events in the first ten years of follow-up to avoid possible inverse causal relationships.

Please add the proportion of excluded missing data. There should be a mentioning, whether missing of data occurred at random.

Please add the description of lost to follow-up.

6. PLOS authors have the option to publish the peer review history of their article (what does this mean?). If published, this will include your full peer review and any attached files.

Reviewer #1: No

Reviewer #2: No

---

## [Author Response · Author response to Decision Letter 0]

19 Nov 2020

Dear Editor:

Please find our revision of manuscript entitled “Joint effect of alcohol drinking and tobacco smoking on all-cause mortality and premature death in China: a cohort study” which resubmit to the journal of PLoS One for consideration.

First of all, we really thanks for the comments from the editors and reviewers which are important and helpful for promoting quality of this paper. The answers of each points of comments from two reviewers was submitted as a word file named “Response to Reviewers”, and requests of editors and journal including format, ethic statement and so on were also been revised in the manuscript accordingly. In addition, we received no specific funding for this work, and state as “The authors received no specific funding for this work.”, hope you could change the online submission form on our behalf. We also confirm that the authors of the present study had no special access privileges in accessing these datasets which other interested researchers would not have. The Supplementary table 2 specified the all relevant data set names, variables used in this paper.

If you need any more information concerning the manuscript please write to me. I am looking forward to your response.

Sincerely yours.

---

## [Decision Letter · Decision Letter 1]

14 Dec 2020

PONE-D-20-20926R1

Joint effect of alcohol drinking and tobacco smoking on all-cause mortality and premature death in China: a cohort study

PLOS ONE

Dear Dr. Yan,

Thank you for submitting your manuscript to PLOS ONE. After careful consideration, we feel that it has merit but does not fully meet PLOS ONE’s publication criteria as it currently stands. Therefore, we invite you to submit a revised version of the manuscript that addresses the points raised during the review process.

We look forward to receiving your revised manuscript.

Kind regards,

Muhammad Aziz Rahman, MBBS, MPH, CertGTC, GCHECTL, PhD

Academic Editor

PLOS ONE

Reviewers' comments:

Reviewer's Responses to Questions

**Comments to the Author**

1. If the authors have adequately addressed your comments raised in a previous round of review and you feel that this manuscript is now acceptable for publication, you may indicate that here to bypass the “Comments to the Author” section, enter your conflict of interest statement in the “Confidential to Editor” section, and submit your "Accept" recommendation.

Reviewer #1: All comments have been addressed

Reviewer #2: All comments have been addressed

2. Is the manuscript technically sound, and do the data support the conclusions?

Reviewer #1: Yes

Reviewer #2: Yes

3. Has the statistical analysis been performed appropriately and rigorously? 

Reviewer #1: I Don't Know

Reviewer #2: Yes

4. Have the authors made all data underlying the findings in their manuscript fully available?

Reviewer #1: Yes

Reviewer #2: Yes

5. Is the manuscript presented in an intelligible fashion and written in standard English?

Reviewer #1: Yes

Reviewer #2: Yes

6. Review Comments to the Author

Reviewer #1: This manuscript is much improved. My comments have been addressed. I would suggest a final edit, to ensure that all terminology matches throughout. For example sometimes cigarette smoking is used and sometimes tobacco smoking is used. I would make these consistent.

Reviewer #2: The authors addressed my comments. The paper will be improved if the tables may be made the figures for visualization. Please refer the Figure 2 and Figure 4 in the paper [https://doi.org/10.1007/s00125-020-05214-4 ]. I have no further concerns.

7. PLOS authors have the option to publish the peer review history of their article (what does this mean?). If published, this will include your full peer review and any attached files.

Reviewer #1: No

Reviewer #2: No

---

## [Author Response · Author response to Decision Letter 1]

18 Dec 2020

Reviewer #1:

Q1: This manuscript is much improved. My comments have been addressed. I would suggest a final edit, to ensure that all terminology matches throughout. For example sometimes cigarette smoking is used and sometimes tobacco smoking is used. I would make these consistent.

Answer: Thanks, we have checked the manuscript carefully, and changed cigarette smoking and alcohol consumption to tobacco smoking and alcohol drinking respectively.

Reviewer #2:

Q1: The authors addressed my comments. The paper will be improved if the tables may be made the figures for visualization. Please refer the Figure 2 and Figure 4 in the paper [https://doi.org/10.1007/s00125-020-05214-4 ]. I have no further concerns.

Answer: Thank you for constructive suggestion. We have draw four figures according to the original table2 and table3 to make results visualized. But we still thought original table 2 and table 3 is valuable for detailed information, therefore, we showed the original table 2 and table 3 in the appendix (S1 table and S2 table).

---

## [Editor Report · Decision Letter 2]

6 Jan 2021

Joint effect of alcohol drinking and tobacco smoking on all-cause mortality and premature death in China: a cohort study

PONE-D-20-20926R2

Dear Dr. Yan,

We’re pleased to inform you that your manuscript has been judged scientifically suitable for publication and will be formally accepted for publication once it meets all outstanding technical requirements.

Kind regards,

Muhammad Aziz Rahman, MBBS, MPH, CertGTC, GCHECTL, PhD

Academic Editor

PLOS ONE

---

## [Editor Report · Acceptance letter]

18 Jan 2021

PONE-D-20-20926R2 

Joint effect of alcohol drinking and tobacco smoking on all-cause mortality and premature death in China: a cohort study 

Dear Dr. Yan:

I'm pleased to inform you that your manuscript has been deemed suitable for publication in PLOS ONE. Congratulations! Your manuscript is now with our production department. 

Kind regards, 

on behalf of

Associate Professor Dr. Muhammad Aziz Rahman 

Academic Editor

PLOS ONE